# The Role of Perivascular Adipose Tissue in the Pathogenesis of Endothelial Dysfunction in Cardiovascular Diseases and Type 2 Diabetes Mellitus

**DOI:** 10.3390/biomedicines11113006

**Published:** 2023-11-09

**Authors:** Alessia Valentini, Carmine Cardillo, David Della Morte, Manfredi Tesauro

**Affiliations:** 1Department of Systems Medicine, Tor Vergata University, 00133 Rome, Italy; alessiasara85@tiscali.it (A.V.); david.dellamorte@uniroma2.it (D.D.M.); 2Department of Aging, Policlinico A. Gemelli IRCCS, 00168 Roma, Italy; carmine.cardillo@unicatt.it; 3Department of Translational Medicine and Surgery, Catholic University, 00168 Rome, Italy

**Keywords:** non-communicable diseases (NCDs), perivascular adipose tissue (PVAT), endothelial dysfunction (ED), cardiovascular diseases (CVDs), insulin resistance (IR), type 2 diabetes mellitus (T2DM)

## Abstract

Cardiovascular diseases (CVDs) and type 2 diabetes mellitus (T2DM) are two of the four major chronic non-communicable diseases (NCDs) representing the leading cause of death worldwide. Several studies demonstrate that endothelial dysfunction (ED) plays a central role in the pathogenesis of these chronic diseases. Although it is well known that systemic chronic inflammation and oxidative stress are primarily involved in the development of ED, recent studies have shown that perivascular adipose tissue (PVAT) is implicated in its pathogenesis, also contributing to the progression of atherosclerosis and to insulin resistance (IR). In this review, we describe the relationship between PVAT and ED, and we also analyse the role of PVAT in the pathogenesis of CVDs and T2DM, further assessing its potential therapeutic target with the aim of restoring normal ED and reducing global cardiovascular risk.

## 1. Introduction

Non-communicable diseases (NCDs) are chronic medical conditions characterized by long duration and slow progression [1,2,3], resulting from a combination of genetic, physiological, environmental, and behavioural factors [3]. They represent the leading cause of death worldwide [4], being responsible for 90% of deaths and 85% of disability in European countries [5].

The four major NCDs are cardiovascular diseases (CVDs), cancer, chronic respiratory diseases, and diabetes [4,5]. According to data reported by the World Health Organization (WHO) in 2016, CVDs account for about 44% of deaths related to NCDs [4], representing the leading cause of morbidity and mortality among these chronic diseases [4,6,7], whereas diabetes accounts for about 4% of all NCDs deaths [4]. Moreover, T2DM itself is associated with higher incidence of atherosclerotic cardiovascular diseases like coronary artery disease, stroke, and peripheral vascular disease, which are the leading cause of morbidity and mortality in diabetic patients [8,9].

Several risk factors contribute to the pathogenesis of NCDs, the most common of which is obesity, which is itself particularly linked to the increased risk of CVDs and type 2 diabetes mellitus (T2DM) [5].

Based on Word Health Organization criteria, obesity is a chronic disease characterized by a body mass index (BMI) above 30 Kg/m^2^, representing the major modifiable risk factor for metabolic comorbidities, including T2DM, and for cardiovascular diseases, including hypertension, coronary artery disease, and heart failure with preserved ejection fraction [10,11,12,13,14]. From a metabolic point of view, the expansion of adipose tissue observed in obesity is associated with an increased release of inflammatory cytokines and free fatty acids, and with a dysregulated secretion of adipokines, which impair insulin signalling, inducing insulin resistance and thus hyperglycaemia, and also contributing to β-cells failure [15,16]. Furthermore, obesity can directly induce haemodynamic alterations by increasing blood volume and thus inducing left ventricular concentric remodelling, leading to progressive left ventricular dysfunction and to the development of heart failure with preserved ejection fraction (HFpEF) [13,14]. However, obesity may also indirectly contribute to the development of cardiovascular diseases by altering the metabolic profile and then promoting the development of atherosclerosis [13,17].

Therefore, as is well known, obesity is per se associated with altered endothelium-dependent vascular reactivity, even in metabolically healthy obesity, but the presence of metabolic abnormalities contributes to enhance worsening of the endothelial dysfunction (ED) [18,19], which represents the common feature of cardiovascular diseases (CVDs) and type 2 diabetes mellitus (T2DM) [20]. Endothelial dysfunction (ED) is defined as the lack of the vasoprotective homeostatic function of the endothelium [20,21,22] and is characterized by vasoconstriction and pro-thrombotic and pro-inflammatory state [22,23].

Although the first description of endothelial dysfunction (ED) was articulated in the mid-1980s [20,23], subsequent research has demonstrated that systemic oxidative stress is the major mechanism involved in its pathogenesis [23,24,25]. However, recent studies have shown that perivascular adipose tissue (PVAT) also contributes to the development of ED [26,27] by producing a variety of bioactive molecules, like bioactive products of adipose tissue, involved in the regulation of vascular tone and local inflammation [26,28], thus promoting the development and progression of atherosclerosis and CVDs [29,30]. Moreover, some of these adipokines affect insulin sensitivity, contributing to the pathogenesis of T2DM and also to the worsening of ED in this chronic disease [28,29,31].

In this review, we describe the relationship between perivascular adipose tissue (PVAT) and endothelial dysfunction (ED), and we analyse the role of PVAT in the pathogenesis of cardiovascular diseases (CVDs) and type 2 diabetes mellitus (T2DM), further evaluating PVAT as a potential therapeutic target with the aim of restoring normal endothelial function and reducing global cardiovascular risk.

## 2. Endothelial Dysfunction (ED)

The endothelium, a continuous inner monolayer of blood vessels, was initially described as a selectively permeable membrane, separating the vascular and the interstitial compartments and regulating the transport of fluids and macromolecules between these two different sections [24,32,33,34]. Notwithstanding, endothelium is currently known as playing a central role in the modulation of vascular homeostasis [35,36] by producing and releasing a wide range of vasoactive substances, implicated in both vasodilation and vasoconstriction [34,35,36]. Moreover, endothelial cells secrete a variety of molecules involved in the regulation of blood coagulation, platelet function, inflammation, and smooth muscle cells proliferation [34,36].

The key mediator in the maintenance of vascular homeostasis by modulating vascular tone and inhibiting inflammation, thrombosis, and cellular proliferation is nitric oxide (NO) [36,37].

Nitric oxide (NO) is a soluble gas synthetized from L-arginine by the endothelial nitric oxide synthase (eNOS) enzyme, constitutively expressed in the endothelial cells [37].

The reduced bioavailability of NO, observed in the presence of risk factors such as smoking, aging, hypertension, hyperglycaemia, hypercholesterolemia, and obesity [22,33,38], induces the development of endothelial dysfunction (ED) [33,39].

Oxidative stress, characterized by higher production and decreased degradation of reactive oxygen species (ROS), plays a pivotal role in the pathogenesis of ED [23,40] by mediating the endothelial production of cytokines, which in turn contribute to the development and progression of atherosclerotic lesions secondary to the endothelial activation [25,36,41].

Although endothelium has been described as the main regulator of vascular homeostasis, recent studies showed that the perivascular adipose tissue (PVAT) is also an active component of the vasculature [42,43], being involved in the maintenance of vascular function and contributing to the prevention of vascular inflammation and atherosclerosis in healthy condition [44,45].

## 3. Perivascular Adipose Tissue (PVAT)

Perivascular adipose tissue (PVAT) refers to adipose tissue surrounding blood vessels, with the exception of capillaries and pulmonary and cerebral vasculature, and it is phenotypically different from other adipose tissue depots [46,47].

As known, adipose tissue is usually classified as white adipose tissue (WAT), brown adipose tissue (BAT), or beige adipose tissue [48]. WAT is also classified as subcutaneous or visceral, and it represents the major organ storage of triglycerides [48,49], which are released depending on systemic energy levels [50]. Moreover, white adipose tissue (WAT) acts as an endocrine organ by secreting hormones, cytokines, and chemokines [48,49], including adiponectin, leptin, resistin, lipocalin, omentin, fibroblast growth factor-21 (FGF-21), tumour necrosis factor-α (TNF-α), and a variety of different interleukins (Il-6, IL-1β, IL-8, IL-18) [48,50].

Unlike WAT, which is composed of adipocytes containing a single large lipid droplet and few mitochondria [50], brown adipose tissue (BAT) contains a large amount of mitochondria, and it is thus involved in thermogenesis [49].

As mentioned above, WAT includes visceral adipose tissue, composed of omental adipose tissue, mesenteric adipose tissue, and retroperitoneal adipose tissue [48].

Visceral adipose tissue, particularly mesenteric adipose tissue, shows higher lipoprotein lipase activity, thus inducing higher circulating levels of free fatty acids [51] and consequently contributing to the development of insulin resistance and atherosclerosis [52].

Compared with other adipose tissue depots, PVAT resembles either white adipose tissue or brown adipose tissue, depending on its anatomical localization [47,49]. Accordingly, in contrast to the thoracic perivascular adipose tissue, which is similar to brown adipose tissue (BAT), abdominal periaortic vascular tissue presents similar features to white adipose tissue (WAT) [47].

The close similarity between WAT and abdominal PVAT [49], and more in detail between mesenteric adipose tissue and mesenteric PVAT [53], is not only related to their histological features but also associated with their biological properties. Thus, as visceral adipose tissue, mesenteric PVAT expresses some of the genes involved in lipid metabolism, such as lipoprotein lipase, other than several adipokines and cytokines [47,54].

Perivascular adipose tissue (PVAT) has no anatomical barrier with the external layer of blood vessels wall [53,55], and it directly infiltrates the adventitia in small vessels and microvessels [47,55].

PVAT is mainly composed of adipocytes, including mature adipocytes, pre-adipocytes, and adipocyte stem cells, which are surrounded by fibrous connective tissue, nerves, blood vessels, such as *vasa vasorum*, and stromal cells [46,47,53,55].

Stromal cells include fibroblasts, mesenchymal cells, smooth muscle cells, endothelial precursors cells, pericytes, and immune cells such as macrophages, B, and T lymphocytes [47,53,55]. Stromal cells may contribute to vascular homeostasis by modulating inflammation and vascular proliferation [53,56,57].

Although until recently it was viewed as a simply connective tissue providing mechanical support to adjacent structures [46,58], it is now considered a metabolically active tissue, with specific properties and secretory patterns depending on its localization, involved in the modulation of vascular homeostasis and endothelial function and in the development of insulin resistance (IR) [45,46,59].

### 3.1. PVAT Modulation of Endothelial Dysfunction

As mentioned above, PVAT is tightly associated with the blood vessel wall [60,61], directly in contact with the adventitia [59,62]. This close interaction promotes the crosstalk between PVAT and all the cellular components of the vascular wall [61], including endothelial cells, on which PVAT exerts its effects in endocrine, autocrine, and paracrine manner by secreting a wide range of bioactive molecules like vasoregulators, including PVAT-derived relaxing factors (PVRFs) and PVAT-derived contracting factors (PVCFs), adipokines, including adiponectin and leptin, chemokines and cytokines, ROS, angiotensin peptides, and extracellular vesicles containing microRNA (miRNA) (Figure 1) [42,58,59,60,61,62,63,64]. Moreover, the absence of an anatomical barrier between the external layer of the vascular wall and PVAT may facilitate the diffusion of PVAT-derived factors in the vasculature [47].

The role of PVAT in the modulation of vascular homeostasis was first described by Soltis and Cassis after observing a significant decrease in norepinephrine response of the intact aortic ring in vitro, suggesting that PVAT contributes to the removal of catecholamines [65]. However, subsequent studies showed that the vasodilatory effect of PVAT was independent of adrenergic neuronal reuptake of catecholamines and was mediated by a transferable substance, called adipocyte-derived relaxing factor (ADRF), which is released in a Ca^+^-dependent manner and acts by activating ATP-dependent K^+^ channels of the vascular smooth muscle cells (VSMCs) [66,67,68]. As the chemical nature of ADRF is still unknown and there might be more than one single molecule responsible for this effect [67,69], the term PVAT-derived relaxing factors (PVRFs) was coined to identify several bioactive molecules involved in the PVAT-induced vasodilation [42,69].

In addition to ADRF, other possible PVRFs are nitric oxide (NO), hydrogen sulphide (H_2_S), hydrogen peroxide (H_2_O_2_), angiotensin 1–7 (Ang1–7), and palmitic acid methyl ester (PAME) [64,69,70].

Once reaching the endothelial layer of blood vessels, these molecules modulate vascular tone through an endothelium-dependent mechanism, mediated by the release of endothelial NO, which in turn induces the activation of K^+^ channels, leading to vasodilation, or through an endothelium-independent mechanism [71,72,73,74].

PVAT-induced vasodilation can also be directly mediated by PVAT-derived nitric oxide (NO) [73]. Indeed, similar to the endothelial cells, adipocytes also express eNOS enzyme, which induces the production of vasoactive NO and consequently the modulation of vasodilation, by acting on VSMCs, and endothelial protection, by inhibiting platelet aggregation and leukocyte adhesion [71]. However, in addition to PVRFs, perivascular adipose tissue (PVAT) also produces PVAT-derived contracting factors (PVCFs), including superoxide anion, catecholamines, prostaglandins, angiotensin II (Ang II), and resistin, which contribute to counteract the effects of PVRFs, primarily reducing NO bioavailability and thus inducing vasoconstriction and ED [63,64].

As previously described, aside from PVRFs and PVCFs, PVAT secretes a wide range of adipocytokines, including adipokines (Figure 2) and cytokines, involved in the regulation of vascular tone and in the maintenance of metabolic homeostasis [63]. Among the adipokines, adiponectin is the most abundant molecule produced and released by PVAT [73] and plays a critical role in the modulation of vascular tone through the activation of its receptor, particularly of the adiponectin receptor 1 (AdipoR1), widely expressed on endothelial cells and VSMCs [75]. Upon adiponectin binding to the Adipo1 receptor of the endothelial cells, the adaptor protein APPL1 is recruited to the receptor site, leading to the phosphorylation of AMP-activated protein kinase (AMPK) and then to the activation of the eNOS, promoting NO production [75,76]. As well as adiponectin, leptin contributes to vascular homeostasis through the enhanced NO production by eNOS activation in endothelial cells [77,78].

As mentioned above, PVAT also secretes cytokines, such as tumour necrosis factor-α (TNF-α) and interleukin-6 (IL-6), which reduce NO bioavailability contributing to ED [63]. Indeed, PVAT-derived TNF-α primarily interferes with endothelial function through enhancing ROS production, via activation of nicotinamide dinucleotide phosphate oxidase (NADPH oxidase), leading to NO breakdown and to reduced NO production as the result of eNOS uncoupling [79,80,81]. Moreover, PVAT-derived TNF-α directly activates the c-Jun N-terminal kinase pathway contributing to the increased production of endothelin-1 (ET-1), which in turn promotes ROS production and consequently ED [79]. Similar to TNF-α, IL-6 also promotes ED by inhibiting eNOS expression and activity [82,83], leading to decreased NO bioavailability [84].

Moreover, as for white adipose tissue (WAT), perivascular adipose tissue (PVAT) also expresses microRNAs (miRNAs), non-coding RNAs that can modulate vascular remodelling and insulin sensitivity, acting post-transcriptionally through the regulation of gene expression [64,85,86].

Thus, under physiological condition, PVAT exerts a protective effect on endothelial function, contributing to maintain the balance between PVAT-derived anti-contractile/anti-inflammatory and pro-contractile/pro-inflammatory bioactive molecules and ensuring greater production of adiponectin and NO than TNF-α and IL-6, finally preventing the development of CVDs [87].

### 3.2. PVAT Regulation of Insulin Sensitivity

Insulin is an endocrine peptide hormone involved in glucose homeostasis by binding its specific receptors expressed on target tissues, such as skeletal muscle [88], which is responsible for most of the insulin-mediated glucose uptake in the post-prandial state [89,90], and which also promotes glycogen synthesis [88].

Apart from its role in inducing myocellular signalling cascade leading to the translocation of glucose transporter 4 (GLUT4) to the plasma membrane, resulting in glucose uptake [88], insulin enhances glucose disposal in the skeletal muscle also by binding its receptors expressed on the endothelial cells of the skeletal muscle microvasculature, thus inducing vasodilation and consequently increased total blood flow to the muscle [91]. Insulin-mediated vasodilation on the skeletal muscle microvasculature depends on the activation of phosphatidyl inositol 3-kinase (PI3-K) pathway, which induces eNOS, increasing NO production [92,93,94] and consequently promoting muscle microvascular dilation, and thus improving muscle glucose uptake [91,92].

As for other vessels, skeletal microvasculature is also surrounded by PVAT [59], involved in the modulation of muscle insulin sensitivity by releasing adipocytokines, which play a critical role in the regulation of insulin-mediated vasodilation [31,95,96].

Among these adipocytokines, adiponectin is widely known to be an insulin sensitizer [97], acting on the skeletal muscle cells by binding Adipo1 receptor and thus activating an intracellular signalling cascade, leading to GLUT4 membrane translocation and glucose uptake [97,98]. Furthermore, adiponectin modulates muscle microvascular recruitment and insulin delivery to skeletal muscle [98] through the direct activation of the AMPK-eNOS pathway and the indirect phosphorylation of protein kinase B (PKB or Akt), which is involved in most of the metabolic actions of insulin [92,99], leading to microvasculature NO-mediated vasodilation and to increased muscle glucose disposal [97,100,101].

Taking into account that PVAT secretes a copious amount of adiponectin and that adiponectin modulates skeletal muscle microvascular perfusion and glucose uptake, it has been demonstrated that adiponectin derived from PVAT surrounding muscle microvasculature regulates insulin-induced vasodilation through the activation of the AMPK pathway in the microvessels wall [102,103], emphasizing the close interaction between PVAT and muscle microvasculature, and then the pivotal role of PVAT in ensuring glucose homeostasis and insulin sensitivity.

Furthermore, PVAT is able to release Ang1-7, which acts not only as an endothelium-dependent relaxing factor, as previously described, but also as an insulin sensitizer by binding and activating the G-protein-coupled receptor Mas, thus inducing the phosphorylation of the insulin receptor substrate-1 (IRS-1) and consequently the activation of the Akt pathway, which mediates insulin metabolic action [104]. Importantly, by activating Mas in the skeletal muscle microvasculature, Ang1-7 also increases the endothelial surface area available for glucose exchange, leading to increased muscle glucose disposal and enhanced metabolic action of insulin [105,106].

Finally, perivascular adipose tissue (PVAT) releases apelin [47], a bioactive peptide that induces endothelium-dependent vasodilatation by promoting nitric oxide (NO) release through the activation of eNOS in the endothelial cells and in skeletal muscle microvasculature, thus favouring glucose disposal [107]. However, apelin also improves glucose uptake by modulating the insulin pathway through the phosphorylation of Akt and the activation of AMPK [107].

Thus, similar to adiponectin, PVAT-derived Ang1-7 and apelin could be involved in the regulation of glucose metabolism, improving insulin sensitivity.

## 4. Dysfunctional PVAT and Its Implication in CVDs and T2DM

Under physiological conditions, PVAT exerts its protective role on vascular and glucose homeostasis by releasing a wide range of bioactive molecules with anti-contractile and anti-inflammatory effects and by secreting adipocytokines involved in the regulation of insulin sensitivity, thus preventing the development of atherosclerosis and insulin resistance (IR) [26,29,53,108]. In contrast, under pathological conditions, such as obesity, PVAT becomes dysfunctional, contributing to the development of endothelial dysfunction (ED), the hallmark of atherosclerosis and CVDs, and of IR, the hallmark of T2DM (Figure 3) [42,109,110].

### 4.1. Dysfunctional PVAT in the Pathogenesis of Atherosclerosis

The development of PVAT dysfunction in obesity is related to the “obesity triad”, characterized by hypoxia, oxidative stress, and inflammation [111]. Hypoxia seems to be a direct consequence of adipocyte hypertrophy, which in turn is responsible for a higher distance between adipocytes and blood vessels, leading to reduced oxygen diffusion within tissues and to enhanced production of ROS and inflammatory cytokines [60,112,113]. ROS, such as superoxide anion and H_2_O_2_, primarily derived from Nox family of NADPH oxidase and eNOS uncoupling, contribute to the development of ED by reducing the bioavailability of nitric oxide (NO) [114,115]. Moreover, superoxide anion stimulates endothelial production of endothelin-1 (ET-1) [116], which in turn reduces NO bioavailability by up-regulation of caveolin-1 expression, leading to eNOS inhibition, or by increasing NO degradation, dependent on eNOS uncoupling [117]. In addition, ROS may induce the expression of TNF-α [111,118], which is also secreted in large amounts by dysfunctional PVAT [53]. TNF-α contributes to the development of ED by stimulating ROS production through the activation of NADPH oxidase [81,118], by inducing the up-regulation of the c-Jun N-terminal kinase pathway (JNK) unbalancing ET1/NO system in favour of ET-1 [80], and by reducing adiponectin release [119], leading to reduced endothelial NO bioavailability [80,81,118]. At last, PVAT dysfunction is also characterized by hyperleptinemia, which leads to endothelial leptin resistance with a loss of balance between the ET-1/NO system, contributing to decreased NO bioavailability in the vasculature [111,120].

Thus, taken together, these data suggest that dysfunctional PVAT, similar to dysfunctional endothelium, is characterized by decreased secretion of protective factors, like adiponectin and NO, by increased production of ROS, and by the development of a low grade of inflammation, which contributes to a detrimental release of adipokines and cytokines, such as IL-6 and TNF-α [114,121]. Such changes in the secretory pattern of dysfunctional PVAT contribute to the development of ED, which in turn leads to the development of atherosclerotic lesions by increasing the endothelial expression of adhesion molecules, like monocyte chemoattractant protein-1 (MCP-1), vascular cell adhesion molecule-1 (VCAM-1), and intracellular adhesion molecule-1 (ICAM-1), involved in the adherence and migration of monocytes into the subendothelial layer of the intima, where they become macrophages and secrete pro-inflammatory cytokines, inducing low-density lipoprotein (LDL) oxidation, leading to the development and progression of atherosclerotic plaque [20,41,109,122], and thus contributing to the pathogenesis of CVDs [29,122].

Furthermore, dysfunctional perivascular adipose tissue (PVAT) contributes to the development of cardiovascular diseases (CVDs), such as heart failure with preserved ejection fraction (HFpEF), by inducing microvascular dysfunction, particularly coronary microvascular dysfunction [123,124,125]. Coronary microvascular dysfunction, resulting from endothelial dysfunction (ED), is defined as the inability of the coronary artery to increase blood flow in response to stressors in the absence of coronary atherosclerosis [126]. Although the pathophysiology of HFpEF needs further investigation, coronary microvascular dysfunction has been identified as the cause and the maintenance mechanism of this condition by inducing both heightened diastolic stiffness and subendocardial ischemia, which in turn contribute to myocardial fibrosis [126,127].

At least, microvascular alterations induced by dysfunctional PVAT contribute to the development of arterial hypertension [125]. As described above, in healthy subjects, perivascular adipose tissue (PVAT) modulates vascular homeostasis by secreting both vasodilator and vasoconstrictor molecules, the latter of which are mainly represented by endotlein-1 (ET-1) and angiotensin II (Ang II) [128]. Under pathological conditions, such as obesity, the dysfunctional PVAT releases a larger amount of Ang II, leading to an abnormal activation of the renin–angiotensin–aldosterone system, thus promoting endothelial dysfunction (ED) and microvascular remodelling [125,128,129]. Moreover, perivascular adipose tissue (PVAT) dysfunction is associated with an overactivation of the local PVAT sympathetic nervous system and with reduced production of anticontractile adipokines, thus promoting the onset of hypertension [129].

### 4.2. Dysfunctional PVAT Inducing Insulin Resistance

Similar to its contribution to the pathogenesis of CVDs, as previously described, PVAT dysfunction is tightly related to the development of vascular IR [26,29], characterized by the loss of insulin-mediated NO production and consequently by impaired skeletal muscle perfusion contributing to decreased glucose disposal [130]. Indeed, the enhanced production of pro-inflammatory cytokines, particularly TNF-α, which inactivate the Akt/PI3-K/eNOS signalling pathway by increasing the enzyme phosphatase and tensin homologue (PTEN) phosphorylation [131] and by activating the JNK pathway, which induces increased ET-1 production [132], leads to decreased glucose uptake by impairing insulin-mediated vasodilation and reducing GLUT-4 translocation in skeletal muscle [28,31,103,109,133].

Furthermore, it has been shown that PVAT-derived adiponectin modulates insulin-dependent muscle perfusion and glucose uptake [110]. Thus, the decreased release of adiponectin by dysfunctional PVAT contributes to the development of vascular IR by impairing of the insulin-dependent Akt signalling pathway [102].

Accordingly, these data suggest that PVAT plays a central role in the pathogenesis of T2DM by regulating muscle perfusion, insulin sensitivity, and glucose disposal.

## 5. PVAT: New Therapeutic Target?

Dysfunctional PVAT is common in obesity and contributes to the development of ED and IR, which represent the hallmarks of CVDs and T2DM, respectively, as described above. Hence, given the association between PVAT dysfunction and the occurrence of cardiometabolic diseases, restoring PVAT function could contribute to reduce the onset and the progression of these diseases, mainly in obesity.

Two studies have shown that weight loss leads to restoring PVAT function in obesity [134,135]. Particularly, the first study demonstrated that PVAT function was restored six months after bariatric surgery, leading to improvement in local adiponectin and NO bioavailability [134]. However, bariatric surgery is not always the first line of treatment for obesity; thus, subsequent studies analyse the effects of caloric restriction and physical activity on PVAT function in obese mice and demonstrate that diet-induced sustained weight loss and exercise also contribute to improving PVAT function by reducing PVAT inflammation and increasing PVAT–adiponectin availability and PVAT–eNOS activity [135,136], suggesting that restoring PVAT function is associated with improvement in insulin sensitivity and endothelial function. Although further investigations are needed to confirm these results in human obesity, we can speculate that PVAT could be a new therapeutic target, and restoring its function could improve cardiometabolic risk. Consistent with this hypothesis, several studies have analysed the effects of some drugs, such as statins and antidiabetic agents, on PVAT function.

### 5.1. Statins

Despite their beneficial effects in reducing cardiovascular risk through the inhibition of hydroxyl-methyl-glutaryl coenzyme A reductase (HMG-CoA-reductase), which induces LDL reduction [137], it has been described that statins have several pleiotropic effects, some of which are responsible for the endothelial modulation of the Akt pathway and caveolin-1 expression, leading to increased nitric oxide (NO) production and thus improving endothelial dysfunction (ED) [138,139] without affecting insulin resistance (IR) [140,141]. Consistent with these data, it could be hypothesized that statins may mediate PVAT secretion of bioactive molecules involved in the regulation of vascular homeostasis. Accordingly, some studies have demonstrated that atorvastatin could restore PVAT function in rat models by increasing the release of PVRFs [142], like hydrogen sulphide (H_2_S) [143]. To date, further investigations are needed to prove the effects of statins in human PVAT.

### 5.2. Antidiabetic Drugs

Similar to statins, in addition to their glucose-lowering effects, new antidiabetic drugs, especially GLP-1 receptor agonists (GLP-1 RAs) and sodium–glucose cotransporter-2 (SGLT-2) inhibitors, directly reduce cardiovascular risk [144,145] by improving endothelial cell function and reducing inflammation and oxidative stress, and thereby preventing the development and progression of atherosclerosis [146,147,148].

Since PVAT appears to play a central role in the development of atherosclerosis, by contributing to the onset of ED, as previously mentioned, several studies have evaluated the effects of GLP-1 RAs and SGLT-2 inhibitors on dysfunctional PVAT. In this context, it has been demonstrated that the GLP-1 RA liraglutide reduces PVAT inflammation through the inhibition of the nuclear factor (NF)-κ B signalling pathway in rat models [149] and improves ED through the activation of the PVAT-AMPK/eNOS pathway and through the enhancement of PVAT-derived adiponectin bioavailability in mice [150]. In view of these findings, it may be speculated that GLP-1 RAs contribute to restore the bioavailability of adiponectin and NO in PVAT, leading to reduced cardiovascular risk.

Furthermore, treating apolipoprotein-E-deficient (ApoE^−/−^) mice with the SGLT-2 inhibitor empagliflozin was associated with decreased expression of pro-inflammatory cytokines, including adhesion molecules, and with reduced activity of NADPH oxidase in PVAT [151]. Therefore, as well as in animal models, in humans, empagliflozin could also prevent ED and atherosclerosis by reducing PVAT inflammation and oxidative stress, but further studies are needed.

### 5.3. Pharmacological Modulation of AMP-Activated Protein Kinase (AMPK)

As described above, perivascular adipose tissue (PVAT) is implicated in the regulation of vascular homeostasis and insulin sensitivity, also by modulating the function of AMP-activated protein kinase (AMPK) [152].

AMPK is a heterodimeric protein involved in the maintenance of insulin sensitivity and in the modulation of vascular homeostasis, by improving endothelial function [152,153], and its inactivation may contribute to the pathogenesis of cardiovascular diseases (CVDs) and type 2 diabetes mellitus (T2DM) [153].

Since dysfunctional PVAT is associated with reduced activity of AMPK, the pharmacological activation of AMPK pathway may reverse PVAT dysfunction, leading to the prevention and treatment of metabolic and cardiovascular diseases [152,153].

Antidiabetic drugs such as metformin and thiazolidinediones [152,154], other than liraglutide [149], may activate the AMPK pathway, thus promoting glucose transport and increasing nitric oxide (NO) bioavailability [152]. Moreover, other molecules implicated in AMPK activation and consequently associated with the amelioration of PVAT function are salicylate [154], methotrexate [155], resveratrol [154], the isoflavonoid calycosin [156], diosgenin [157], and mangiferin [158].

However, further investigations are needed to better understand the efficacy of these drugs in human PVAT.

## 6. Discussion

PVAT is a metabolically active endocrine/paracrine organ involved in the regulation of vascular homeostasis and the impact on insulin sensitivity. The loss of balance between anti-contractile/anti-inflammatory and pro-contractile/pro-inflammatory adipocytokines, as observed in obesity, characterizes PVAT dysfunction, which contributes to the pathogenesis of both CVDs and T2DM. Therefore, restoring PVAT function could improve cardiometabolic risk, leading to a decreased onset of ED and IR. Although caloric restriction and physical activity contribute to restore PVAT function, several studies have investigated the role of statins and antidiabetic drugs on dysfunctional PVAT in animal models, showing that they could modulate several intracellular signalling pathways and regulate PVAT secretion of adipocytokines, thus contributing to prevent ED and atherosclerosis. Regarding their effects on insulin-mediated vasodilation, which also depends on PVAT function, the literature evidence is scarce.

In conclusion, dysfunctional PVAT could be a new therapeutic target for treatment and prevention of CVDs and potentially of T2DM.

## Figures and Tables

**Figure 1 biomedicines-11-03006-f001:**
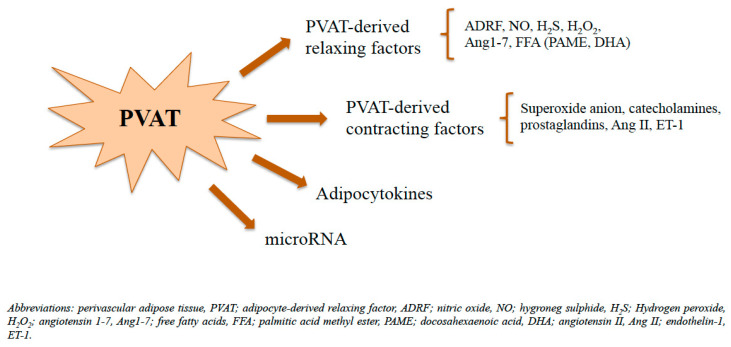
PVAT-derived bioactive molecules.

**Figure 2 biomedicines-11-03006-f002:**
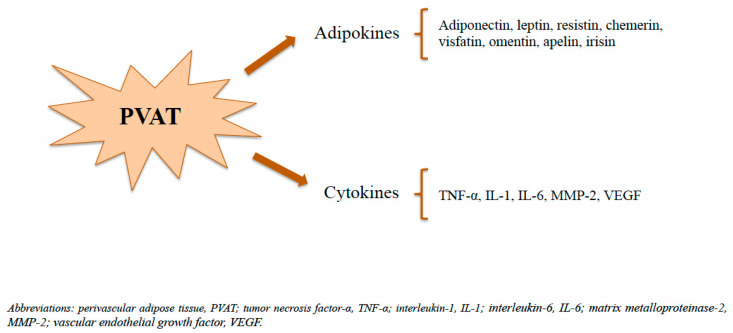
PVAT-derived adipokines and cytokines.

**Figure 3 biomedicines-11-03006-f003:**
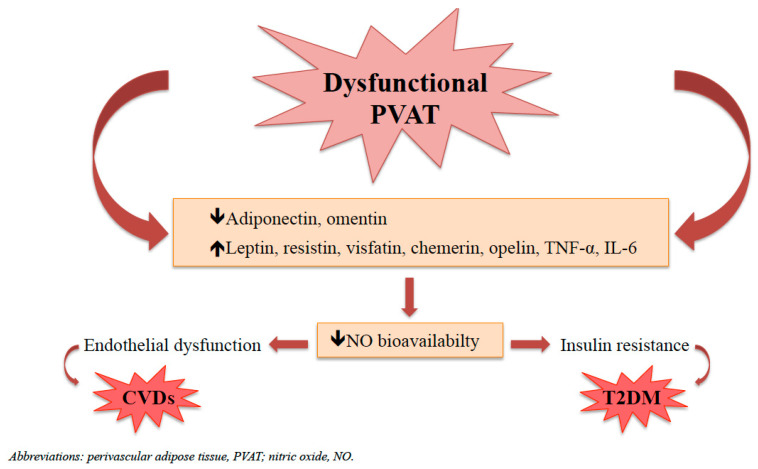
The relationship between dysfunctional PVAT, cardiovascular diseases (CVDs) and type 2 diabetes mellitus (T2DM).

## Data Availability

Data sharing is not applicable to this article.

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
