# Peer review of "The Role of Perivascular Adipose Tissue in the Pathogenesis of Endothelial Dysfunction in Cardiovascular Diseases and Type 2 Diabetes Mellitus"

_biomedicines, 2023, doi:10.3390/biomedicines11113006_

Round 1
Reviewer 1 Report
Comments and Suggestions for Authors
To the authors:
This is a nice comprehensive review on perivascular adipose tissue, which receives increasing attention as a site of vascular inflammation and regulation of vascular function and dysfunction. The storyline is clear although similar to other reviews on perivascular adipose tissue, the information provides could be expanded in my opinion. Consequently I have a couple of suggestions for revising and improving the review:
1. An illustration with mechanisms linking PVAT to endothelial dysfunction, diabetes and vascular disease would be valuable and enhance readability. These illustrations also increase citations as they are often used in presentations.
2. A risk in the PVAT is preaching to the choir, i.e. assuming the importance of PVAT without discussing the role of PVAT in the adipose system. How do local PVAT depots relate to circulating adipokines and e.g. mesenteric fat tissue?
3. The best known adipokines produced by PVAT are now mentioned, could the authors provide a more complete list of known PVAT products?
4. The cellular composition of PVAT (l.85 "stromal cells") is not discussed in detail, but is a known determinant of PVAT function and as such, it should be described and discussed in a separate section.
5. What is known about the molecular determinants of PVAT function, e.g. miRNAs and other epigenetic mechanisms?
6. Regarding the role of PVAT in CVD, the review now zooms in on atherosclerosis but hypertrophy of PVAT also associates with microvascular CVDs such as hypertension and HFpEF. As PVAT and microvascular dysfunction are connected, I would mention and discuss this given the theme of the review.
Minor comments:
7. l.34 the 4% deaths from diabetes is true, as is the association of diabetes with CVD; diabetes likely increases mortality through is sequelae and this is worth mentioning.
8. l.48 the distinction between bioactive molecules released by PVAT and adipokines is not clear. I would define adipokines as bioactive products of adipose tissue.
9. l.68 "arginina" should be "arginine".
10. l.93 "direct contact with the adventitia" what is the authors' view of the functional relationship between the two layers? as they are in direct contact this is worth discussing.
11. l.130 the adiponectin receptor is usually abbreviated AdipoR, with subtypes AdipoR1 and AdipoR2.
12. ll.156-165 the vascular contribution to glucose uptake has bene beautifully shown by David Wasserman's group, and by the group of Kubota et al.
13. l.180 Thank you for referring to our Diabetes paper, but PVAT control of glucose uptake likely involves more than adiponectin, especially in insulin resistance and type 2 diabetes
Author Response
We thank the Reviewer for his/her appreciation, for careful revision and comments.
- An illustration with mechanisms linking PVAT to endothelial dysfunction, diabetes and vascular disease would be valuable and enhance readability. These illustrations also increase citations as they are often used in presentations.
We thank the Reviewer for his/her suggestion. We have added Figure 3.
- A risk in the PVAT is preaching to the choir, i.e. assuming the importance of PVAT without discussing the role of PVAT in the adipose system. How do local PVAT depots relate to circulating adipokines and e.g. mesenteric fat tissue?
The role of PVAT in the adipose tissue and the relation with the mesenteric fat tissue have been discussed from line 122 to line 158.
- The best known adipokines produced by PVAT are now mentioned, could the authors provide a more complete list of known PVAT products?
A more complete list of know PVAT products is reported in Figure 1 and 2
- The cellular composition of PVAT (l.85 "stromal cells") is not discussed in detail, but is a known determinant of PVAT function and as such, it should be described and discussed in a separate section.
The cellular composition of PVAT has been reviewed and discussed from line 159 to line 170.
- What is known about the molecular determinants of PVAT function, e.g. miRNAs and other epigenetic mechanisms?
miRNA are introduced in lines 186-187 and they are described from line 265 to line 269
- Regarding the role of PVAT in CVD, the review now zooms in on atherosclerosis but hypertrophy of PVAT also associates with microvascular CVDs such as hypertension and HFpEF. As PVAT and microvascular dysfunction are connected, I would mention and discuss this given the theme of the review.
The association with hypertension and HFpEF has benne reviewed ad added from line 394 to line 418.
Minor comments:
- l.34 the 4% deaths from diabetes is true, as is the association of diabetes with CVD; diabetes likely increases mortality through is sequelae and this is worth mentioning.
The increased mortality related to diabetes complications has been reported from line 36 to line 40.
- l.48 the distinction between bioactive molecules released by PVAT and adipokines is not clear. I would define adipokines as bioactive products of adipose tissue.
The term “adipokines” has been modified as “bioactive products of adipose tissue”
- l.68 "arginina" should be "arginine".
We apologize with the reviewer for this typo. It has been corrected.
- l.93 "direct contact with the adventitia" what is the authors' view of the functional relationship between the two layers? as they are in direct contact this is worth discussing.
It has been discussed from line 187 to line 190.
- l.130 the adiponectin receptor is usually abbreviated AdipoR, with subtypes AdipoR1 and AdipoR2.
We are indebted to the Reviewer for having noticed this typo, which have been corrected in the revised manuscript.
- ll.156-165 the vascular contribution to glucose uptake has been beautifully shown by David Wasserman's group, and by the group of Kubota et al.
We thank the Reviewer for his/her suggestion. We added the references suggested:
- Kubota T, Kubota N. The role of endothelial insulin signaling in the regulation of glucose metabolism. Rev Endocr Metab Disord 2013, 14 (2), 207-216;
- Kubota T, Kubota N. Impaired insulin signaling in endothelial cells reduces insulin-induced glucose uptake by skeletal muscle. Cell Metab 2011, 13 (3), 294-307;
- Williams IM, Wasserman DH. Capillary endothelial insulin transport: the rate-limiting step for insulin-stimulated glucose uptake. Endocrinology 2022, 163 (2), bqab252.
- l.180 Thank you for referring to our Diabetes paper, but PVAT control of glucose uptake likely involves more than adiponectin, especially in insulin resistance and type 2 diabetes.
We thank the Reviewer for his/her clarification. We have discussed (e.g.) the role of apelin (lines 325-331).
Reviewer 2 Report
Comments and Suggestions for Authors
Dear Authors,
This is interesting paper concerned important problem of perivascular adipose tissue impact on endothelial dysfunction and insulin resistance, what is very important in type 2 diabetes. But I want to underline a few comments:
1. The main suggestion is that Authors should revise not only statins and GLP-1, but also other antidiabetic drugs, especially metformin and thializidinedions.
2. In Pubmed there are other potential substances which probably influence on this adipose- it should be reviewed
3. Should be mentioned about positive impact of pharmacological activation of AMPK on PVAT.
4. Manuscript is written in language very difficult to read. Multi compound sentences are logic and very comprehensive but very difficult to read, like sentence in 5-6 lines: e.g. lines 58-62, or 70-76, 44-50
5. The amount of abbreviations and acronims makes it difficult to read. They should be explained more frequently or if abbreviation will be rarely used, delete.
6. Lines 259-262- two studies are not several, and “previous” better replace by Authors name or “first of referred” . Or maybe better will be” Particularly, the first study , demonstrated …..
Comments on the Quality of English LanguageSuggestions in comments to Authors
Author Response
We thank the Reviewer for his/her appreciation, for careful revision and comments.
- The main suggestion is that Authors should revise not only statins and GLP-1, but also other antidiabetic drugs, especially metformin and thializidinedions.
We revised the role of other antidiabetic drugs and we added a new paragraph (lines 518-527).
- In Pubmed there are other potential substances which probably influence on this adipose- it should be reviewed
We are grateful to the Reviewer for this important comment. We have added further information about other substances which influence PVAT (lines 521-527).
- Should be mentioned about positive impact of pharmacological activation of AMPK on PVAT.
The role of activation of AMPK has been explained in the new paragraph added (lines 506-521).
- Manuscript is written in language very difficult to read. Multi compound sentences are logic and very comprehensive but very difficult to read, like sentence in 5-6 lines: e.g. lines 58-62, or 70-76, 44-50.
We apologize with the reviewer for the lack of clarity. In response to this observation we have corrected the sentences suggested by the reviewer and edited the manuscript.
- The amount of abbreviations and acronims makes it difficult to read. They should be explained more frequently or if abbreviation will be rarely used, delete.
We thank the reviewer for the suggestion. We have explained more frequently the abbreviation used and we have deleted those rarely used.
- Lines 259-262- two studies are not several, and “previous” better replace by Authors name or “first of referred” . Or maybe better will be” Particularly, the first study , demonstrated …..
We thank the reviewer for the clarification. In response to this observation we have corrected the sentences suggested by the reviewer and edited the manuscript (lines 448-451).
Reviewer 3 Report
Comments and Suggestions for Authors
This review article examines the function of perivascular adipose tissue (PVAT) in the progression of endothelial dysfunction (ED) related to cardiovascular diseases and type 2 diabetes mellitus (T2DM). It describes the connection between PVAT and ED, the part of PVAT in CVDs and T2DM's pathogenesis, and assesses PVAT as a potential therapeutic target for restoring typical endothelial function and decreasing cardiovascular risk.
The study is intriguing and relevant, yet there is room for improvement.
1. The paper lacks in-depth analysis of the correlation between obesity and the heightened susceptibility to CVDs and type 2 diabetes mellitus. Expanding on this could provide a more comprehensive understanding of the topic.
2. The work would greatly benefit from the inclusion of figures or diagrams to enhance comprehension.
3. Many references in this review are outdated and do not reflect the latest research. A review should aim to provide up-to-date information.
4. Check the references, for example number 13 is not correct.
Author Response
We thank the Reviewer for his/her careful revision and insightful comments, to whom we reply below.
The paper lacks in-depth analysis of the correlation between obesity and the heightened susceptibility to CVDs and type 2 diabetes mellitus. Expanding on this could provide a more comprehensive understanding of the topic.
The correlation between obesity and the heightened susceptibility to CVDs and type 2 diabetes mellitus has been discussed from line 44 to line 60.
- The work would greatly benefit from the inclusion of figures or diagrams to enhance comprehension.
We thank the Reviewer for his/her suggestion. We have added some figures.
- Many references in this review are outdated and do not reflect the latest research. A review should aim to provide up-to-date information.
We are grateful to the Reviewer for this important comment. In response to this observation we have modified the manuscript by inserting more recent references.
- Check the references, for example number 13 is not correct.
We apologize with the Reviewer for the mistake. The reference has been corrected.
Round 2
Reviewer 3 Report
Comments and Suggestions for Authors
My comments have been adequately addressed.